# Different Strategies to Attenuate the Toxic Effects of Zinc Oxide Nanoparticles on Spermatogonia Cells

**DOI:** 10.3390/nano12203561

**Published:** 2022-10-11

**Authors:** Mariana Vassal, Cátia D. Pereira, Filipa Martins, Vera L. M. Silva, Artur M. S. Silva, Ana M. R. Senos, Maria Elisabete V. Costa, Maria de Lourdes Pereira, Sandra Rebelo

**Affiliations:** 1Department of Biology, University of Aveiro, 3810-193 Aveiro, Portugal; 2Institute of Biomedicine—iBiMED, Department of Medical Sciences, University of Aveiro, 3810-193 Aveiro, Portugal; 3LAQV-REQUIMTE, Department of Chemistry, University of Aveiro, 3810-193 Aveiro, Portugal; 4CICECO-Aveiro Institute of Materials, University of Aveiro, 3810-193 Aveiro, Portugal; 5Department of Materials and Ceramic Engineering, University of Aveiro, 3810-193 Aveiro, Portugal; 6Department of Medical Sciences, University of Aveiro, 3810-193 Aveiro, Portugal

**Keywords:** zinc oxide nanoparticles, cytotoxicity, chalcone, antioxidant, oxidative stress, cytoskeleton, DNA damage, male infertility, reversibility

## Abstract

Zinc oxide nanoparticles (ZnO NPs) are one of the most used nanoparticles due to their unique physicochemical and biological properties. There is, however, a growing concern about their negative impact on male reproductive health. Therefore, in the present study, two different strategies were used to evaluate the recovery ability of spermatogonia cells from the first stage of spermatogenesis (GC-1 spg cell line) after being exposed to a cytotoxic concentration of ZnO NPs (20 µg/mL) for two different short time periods, 6 and 12 h. The first strategy was to let the GC-1 cells recover after ZnO NPs exposure in a ZnO NPs-free medium for 4 days. At this phase, cell viability assays were performed to evaluate whether this period was long enough to allow for cell recovery. Exposure to ZnO NPs for 6 h and 12 h induced a decrease in viability of 25% and 41%, respectively. However, the recovery period allowed for an increase in cell viability from 16% to 25% to values as high as 91% and 84%. These results strongly suggest that GC-1 cells recover, but not completely, given that the cell viability does not reach 100%. Additionally, the impact of a synthetic chalcone (*E*)-3-(2,6-dichlorophenyl)-1-(2-hydroxyphenyl)prop-2-en-1-one (**1**) to counteract the reproductive toxicity of ZnO NPs was investigated. Different concentrations of chalcone **1** (0–12.5 µM) were used before and during exposure of GC-1 cells to ZnO NPs to mitigate the damage induced by NPs. The protective ability of this compound was evaluated through viability assays, levels of DNA damage, and cytoskeleton dynamics (evaluating the acetylated α-tubulin and β-actin protein levels). The results indicated that the tested concentrations of chalcone **1** can attenuate the genotoxicity induced by ZnO NPs for shorter exposure periods (6 h). Chalcone **1** supplementation also increased cell viability and stabilized the microtubules. However, the antioxidant potential of this compound remains to be elucidated. In conclusion, this work addressed the main cytotoxic effects of ZnO NPs on a spermatogonia cell line and analyzed two different strategies to mitigate this damage, which represent a significant contribution to the field of male fertility.

## 1. Introduction

A growing amount of evidence has demonstrated that zinc oxide nanoparticles (ZnO NPs) can interfere with all types of male reproductive cells in a dose-and time-dependent manner [1,2,3]. Due to their widespread use in personal care products, coatings, paints, electronic circuits, sensors [4], and even in antimicrobial food packaging [5], its accidental inhalation, ingestion, or dermal absorption is almost inevitable [6].

In this context, it is essential to evaluate not only the immediate effects of ZnO NPs on reproductive cells, but also how these effects evolve over time. To date, there are no studies about the reversibility of the cytotoxic effects of ZnO NPs in a male reproductive cell line, but what happens to cells upon the removal of the NPs must be taken into consideration when assessing the reproductive toxicity of nanomaterials. While there is no consensus on the effects of NPs on the male reproductive system, the search for new approaches to protect male reproductive health is urgent and might be a significant contribution to the field of male fertility.

There is a growing interest in antioxidants as preventative or therapeutic agents against NPs-mediated injuries [7,8,9]. However, few studies have described the toxicity and protective effects of antioxidants against ZnO NPs in cells from the early stages of spermatogenesis. Some studies have indicated that chalcones may have antioxidant properties due to their free radical scavenging activities [10]. Chalcones (1,3-diarylprop-2-en-1-ones) are natural products with a wide distribution in different plant families belonging to the flavonoid group [11,12]. Chalcones are often referred to as ‘open chain flavonoids,’ since they lack the heterocyclic C ring [13]. These compounds have two aromatic rings (rings A and B) that are connected by a three-carbon α,β-unsaturated carbonyl system. Due to their structural simplicity, these molecules can be easily synthesized and decorated with different groups in the aromatic rings so that the resulting products bind to different biomolecules and present distinct properties [11,14]. In fact, chalcones are so versatile and abundant that there are several thousand naturally occurring chalcones reported in the literature [15,16]. These molecules have been explored for thousands of years using plants and herbs for the treatment of different pathologies, such as cancer, inflammation, and diabetes [16]. Researchers are often inspired by the structure of naturally occurring chalcones to synthesize derivatives with enhanced bioactivities and/or reduced toxicity [15]. Within the broad spectrum of therapeutic applications that natural and synthetic chalcones display, their antioxidant potential is the most explored [17]. Chalcones have been the subject of several review manuscripts describing their natural occurrence in plants, their broad and remarkable biological activities, and their methods of synthesis [16,18]. Additionally, several research articles have been published regarding the synthesis of novel chalcones derivatives and the investigation of their pharmacological activities. Two examples among a huge number of articles are, e.g., the study developed by Rocha et al. that synthesized a wide range of chalcone derivatives and evaluated their ability to inhibit the carbohydrate-hydrolyzing enzymes α-amylase and α-glucosidase to be used as antidiabetic agents [19], and that of Martins et al., who synthesized several families of chalcones and investigated their potential as scavengers of HOCl and as inhibitors of oxidative burst [10].

Antioxidants are substances that can protect cells from oxidative damage caused by free radicals and their derivatives. Free radicals, such as superoxide radicals (O_2_^●−^), hydrogen peroxide (H_2_O_2_), hydroxyl radicals (^●^OH), and singlet oxygen (^1^O_2_), are commonly referred to as reactive oxygen species (ROS). These are generated as metabolic by-products, mainly in mitochondria [20]. Although they are important components in cell respiration and other vital cellular processes, they are unstable and highly reactive molecules. They can easily start a cascade of chain reactions, leading to oxidative stress, which can seriously alter cell membranes and structures [21]. Several in vitro and in vivo studies have demonstrated that ZnO NPs induce substantial cytotoxicity, mainly through the overproduction of ROS [3,22,23,24,25]. We might speculate that the increase in ROS is normally accompanied by an increase in DNA damage and consequently, increased genotoxicity.

The ZnO NPs, being one of the most synthesized types of metal oxide NPs, have been used in various commercial products, including medicine, agriculture, pharmaceuticals, and cosmetics [26]. However, their widespread applications result in higher human exposure and new biological interactions [27].

To a certain degree, the cells’ antioxidant defense systems can minimize levels of ROS, compensate for the oxidative stress, and even repair the DNA damage, if necessary. However, when the intracellular antioxidant system is not able to maintain the proper balance of free radicals due to their chronic production, several health problems may occur [20,25]. In fact, oxidative stress is related to many reproductive disorders [28]. It is an emerging risk factor for male infertility, since it can interfere with sperm quality, mainly by inducing DNA and protein oxidation, as well as lipid peroxidation [29]. Therefore, antioxidant therapies have been considered viable treatment options for male infertility since they are able to restore the proper balance between free radicals and antioxidants in the reproductive cells [30]. Some chalcones have been reported to have antioxidant properties [11,15,31,32,33]. This activity is highly influenced by the structure of the two aromatic rings in the backbone [34] and by the number and position of the hydroxy group, methoxy group, and other groups in the A and B rings [35]. Although the antioxidant potential of many chalcone analogues has been found in many cell lines, the protective effects of this compound on male reproductive cells have not been previously studied.

Consequently, the present study aimed to evaluate two different strategies to attenuate or reverse the damage induced by a cytotoxic concentration of ZnO NPs in a spermatogonia cell line (GC-1 spg). First, the effect of ZnO NPs on GC-1 cells was compared at two different timepoints (6 and 12 h), immediately after exposure to ZnO NPs and after a recovery period, to understand if these cells have the required defense mechanisms to recover from the induced damage. Then, we proceeded to evaluate, for the first time, the protective effects of different doses of the synthesized chalcone (*E*)-3-(2,6-dichlorophenyl)-1-(2-hydroxyphenyl)prop-2-en-1-one (**1**) in GC-1 cells. The following outcomes were monitored, namely cell viability, DNA damage, and alterations in cytoskeleton dynamics through evaluation of the acetylated α-tubulin and β-actin protein levels.

## 2. Materials and Methods

### 2.1. Characterization of ZnO NPs

A ZnO nanopowder, supplied by Sigma-Aldrich, Saint Louis, MO, USA, referenced 544906, was used. The powder was characterized in terms of its crystallography, morphology, particle size distribution, and particle surface charge in aqueous suspensions, with a set of techniques that are described below.

For the crystallographic characterization, X-ray diffraction (XRD) using a X’Pert Pro Diffractometer, PANalytical B.V., Almelo, Netherlands, with Cu K_α1_ radiation (λCu = 0.154056 nm) was applied. The morphology of the particles was inspected by scanning electron microscopy (SEM) performed in a Hitachi SU-70 Scanning Electron Microscope, Tokyo, Japan, and the specific surface area of the powder was accessed by gas adsorption (BET isotherm) in a Micromeritics-Gemini V2380 surface area analyzer, Micromeritics, Norcross, GA, USA. The particle size distribution (PSD) was determined in a water medium by laser diffraction through a Coulter LS-200 device, Beckman Coulter, Brea, CA, USA, and the particle surface charge was also accessed by measuring the Zeta potential of ZnO aqueous suspensions, at different pH levels, in a Coulter Delsa 440 SX device, Beckman Coulter, Indianapolis, IN, USA.

### 2.2. Synthesis of (E)-3-(2,6-Dichlorophenyl)-1-(2-hydroxyphenyl)prop-2-en-1-one (***1***)

The (*E*)-3-(2,6-dichlorophenyl)-1-(2-hydroxyphenyl)prop-2-en-1-one (**1**) was synthesized by base-catalyzed Aldol-type condensation of 2′-hydroxyacetophenone with 2,6-dichlorobenzaldehyde (Figure 1). The synthesis and nuclear magnetic resonance (NMR) spectroscopy data of this compound are available in the Appendix A).

Chalcone **1** was dissolved in dimethylsulfoxide (DMSO; PanReac AplliChem) and stored at 4 °C in the dark. Prior to use, it was diluted in a cell culture medium to create the final concentrations (1.6, 3.1, 6.25, 12.5, and 25 µM). The DMSO final concentration did not exceed 0.25% (*v*/*v*).

### 2.3. Cell Culture

The biological effects of ZnO NPs and synthetic chalcone **1** were studied in the mouse-derived spermatogonia cell line, GC-1 spg cells (ATCC^®^, CRL2053™), a cell line that exhibits phenotypic features of mouse type B spermatogonia and early spermatocytes [36]. GC-1 cells were cultured in Dulbecco’s Modified Eagle’s Medium (DMEM, Sigma-Aldrich) containing 10% (*v*/*v*) of Fetal Bovine Serum (FBS, bioWest) and 1% (*v*/*v*) penicillin-streptomycin solution (PenStrep, bioWest), under standard cell culture conditions (humidified incubator under 5% CO_2_, at 37 °C). After reaching 80% of cell confluence, cells were subcultered using trypsin-EDTA (Gibco) in new 100 mm plates (cell culture maintenance) or seeded in 6-well plates (experiments) and allowed to adhere for 16 h before any treatment.

### 2.4. Experimental Design

To investigate whether GC-1 cells could recover from the damage induced by ZnO NPs (Appendix A), cells were seeded at a density of 30,000 cells/well in 6-well plates and after 16 h, were exposed to 20 µg/mL of ZnO NPs for 6 and 12 h. Then, cells were extensively washed with 1x PBS and maintained in an NPs-free culture medium for the next 4 days to evaluate if the cells were able to repair and restore themselves. Essentially, a cell viability assay was performed at two different timepoints: at the end of the exposure times to ZnO NPs and after the 4-day recovery period (Appendix A).

For the second strategy treatment, cells were plated at a density of 250,000 cells/well in 6-well plates and pre-treated with various concentrations of chalcone **1** (1.6, 3.1, 6.25, and 12.5 µM) for 1 h prior exposure to ZnO NPs (20 µg/mL), and both chalcone **1** and ZnO NPs were left for 6 and 12 h (Appendix A). The chalcone **1** concentrations were chosen based on previous studies using these types of compounds as inhibitors of oxidative burst [10]. For further information, please see the concentrations used for chalcone **4b** (same structure as chalcone **1**) in the referred study.

GC-1 cells were also cultured with chalcone **1** (0–25 µM for 7 and 13 h) and ZnO NPs (20 µg/mL for 6 and 12 h) independently to understand how each treatment affects the cells (Appendix A). The control group included cells without any treatment. At least 3 replicates from each group were performed independently.

### 2.5. Cell Viability Assay

For both treatment strategies, the viability of the GC-1 cells was measured using the resazurin reduction assay. Viable cells with an active metabolism contain coenzymes, such as NADH, which are used in diaphorases to reduce resazurin (blue, non-fluorescent) to resorufin (pink, fluorescent) [37]. The level of reduction is proportional to the number of viable cells. Resazurin was chosen to perform a cell viability assay because it is not toxic and, therefore, cells exposed to it can remain in the culture to be used for other experimental purposes [38].

The cells were incubated with 10% of the culture volume with a stock solution of resazurin (Sigma-Aldrich) (0.1 mg/mL) in 1x PBS [39] 4 h before the end of the 6 and 12 h incubation timepoints and 4 h before the end of the 4-day recovery period. After 4 h of incubation, 100 µL of the supernatant from each sample was transferred to a 96-well plate, and the absorbance of resazurin was measured spectrophotometrically at 570 and 600 nm (Infinite M200, PRO, Tecan). All absorbance values were corrected against blank wells containing cell-free culture medium with ZnO NPs and different concentrations of chalcone **1**. Cells were visualized daily, under an inverted light microscope (EVOS™ M5000 Imaging System), to confirm cell confluence and morphology.

### 2.6. SDS-PAGE and Immunoblotting

At the end of the exposure times, protein lysates were prepared from cells that were co-exposed to chalcone **1** and ZnO NPs. This was done by harvesting cells with 1% boiling sodium dodecyl sulfate (SDS) and boiling them at 95 °C for 10 min. Then, lysates were sonicated with a probe sonicator for 10 s (60% amplitude, 0.5 s cycles) [40,41]. The total protein concentration of the cell lysates was determined by Pierce’s bicinchoninic acid (BCA) protein assay kit (Thermo Fisher Scientific), used according to the manufacturer’s instructions, and quantified by an absorbance reader (Infinite M200, PRO, Tecan).

After protein quantification, 30 µg of protein samples were further denatured by boiling for 5 min in the presence of 1% SDS and loading buffer with β-mercaptoethanol. Protein samples were then separated on 5%–20% gradient SDS-PAGE gels, and electrotransferred onto nitrocellulose membranes (0.2 µm pore size; GE Healthcare) [41]. These membranes were then stained with a Ponceau S solution (Sigma-Aldrich) to assure equal protein loading, and the membranes were scanned on a GS-800 calibrated image densitometer (Bio-Rad). Membranes were incubated for 2 h with a blocking solution (3% bovine serum albumin (BSA)/1x TBS-T), and then incubated for 2 h at room temperature, followed by incubation overnight at 4 °C with the following antibodies: anti-γ-H2AX (S139) (Millipore, Darmstadt, Germany; 1:500), anti-α-tubulin-acetylated (Sigma-Aldrich, Saint Louis, MO, USA; 1:2000), anti-β-tubulin (Invitrogen, Thermo Fisher Scientific, Waltham, MA, USA; 1:1000), and anti-β-actin (Novus Biologicals, Centennial, CO, USA; 1:5000). The primary antibody incubation was followed by three washes of 10 min each with TBS-T. Then, all membranes were incubated with horseradish peroxidase-conjugated secondary antibodies (Cell Signalling Technology, Danvers, MA, USA; 1:10,000) for 1 h at room temperature. Following three washes with TBS-T, the membranes were incubated with the ECL Select WB detection reagent (GE-Healthcare), and the protein bands were detected by chemiluminescence on the Chemi-Doc Imaging System, using Quantity One densitometry software (Bio-Rad). For reprobing, blots were stripped in a mild stripping solution, followed by several washes with 1x TBS-T.

### 2.7. Statistical Analysis

Statistical analysis was performed using GraphPad Prism software (version 8.0.1 for Windows, GraphPad Software, Inc., San Diego CA, USA), using a common Student’s *t*-test or one-way ANOVA, followed by Dunnett’s multiple comparison test with a statistical confidence coefficient of 0.95. However, when data did not meet the assumption of normality or homogeneity of variance, the nonparametric Kruskal–Wallis one-way ANOVA test was performed, followed by Dunn’s multiple comparisons test. A minimum of three independent experiments were performed. All data were expressed as mean ± standard error of the mean (SEM).

## 3. Results

### 3.1. Characterization of ZnO NPs

The main characteristics of the ZnO NPs used in the present study will be detailed below. Their crystallographic structure determined by XRD is identified with that of hexagonal wurtzite, group P63mc, characteristic of ZnO. The specific surface area of the powder, determined by BET isotherm, was S_BET_ = 12 m^2^/g, and from this value, an average equivalent to spherical diameter in the nanometric range, G_BET_ = 88 nm, was calculated.

Further, Figure 2 presents the particle size distribution (PSD) determined by laser diffraction, with superimposed SEM micrographs showing the powder morphology. The particle sizes are distributed from ~40 nm up to ~5000 nm within a bimodal distribution, where the first maximum at ~100 nm may correspond to individualized nanosized particles, and the second peak at ~2000 nm is clearly correspondent to agglomerates formed by the nanometric particles, as is evidenced by the SEM superimposed images. By taking the calculated average particle size of 1360 nm from PSD and assuming G_BET_ as representative of the individual grain size, a high agglomeration factor of 15.5 was calculated for this nanometric powder in near-neutral conditions. The Zeta potential measured at pH~7 is negative, −15 mV, although not high enough to establish a predominant repulsive interaction among particles in aqueous suspensions.

To attenuate the toxicity induced by ZnO NPs in spermatogonia cells, two different approaches were used, as shown in Appendix A. The first was based on GC-1 cell recovery for 4 days after removal of ZnO NPs from the cell culture (Section 3.2). The second approach was based on simultaneous incubation with a synthesized chalcone **1** (Section 3.3 and Section 3.4 and Section 3.5).

### 3.2. Effect of a Recovery Period on GC-1 Cells Survival

Bearing in mind previous studies from our team [1], in which GC-1 cells were exposed to high doses of ZnO NPs for short periods of time (6 h and 12 h) and cell toxicity, DNA damage, and dramatic alterations in the cytoskeleton and nucleoskeleton were observed, we hypothesized that it may be possible for the observed toxicity of the GC-1cells to somehow be reversed. To test this hypothesis, two approaches were employed. We investigated the possibility that GC-1 cell toxicity may be reversed by simply removing the ZnO NPs and if chalcone **1** may counteract the observed GC-1 cell damage. The 20 μg/mL dose of ZnO NPs ensures a strong effect on the cells, maintaining cellular mortality at an acceptable level to evaluate cell recovery [1]. The viability of GC-1 cells was determined at the end of these timepoints by the resazurin assay [42]. The viability of GC-1 cells was negatively affected by ZnO NPs, although to varying degrees (Figure 3). Upon 6 h of exposure, there was a viability decrease of 25%, compared to control. However, this difference was not statistically significant. After 12 h, there was a significant decrease of 41% in the cell viability (*p* < 0.01). At the end of these timepoints, the ZnO NPs solution was removed, cells were washed three times with 1x PBS, and a fresh cell culture medium without ZnO NPs was added. Cells were allowed to recover for 4 days. After this period, new viability assays were performed. Cells that were exposed to ZnO NPs for 6 h had a 16% increase in cell viability. Those exposed for 12 h recovered more readily, with a 25% increase (Figure 3). The results indicate that no difference was measured between the treatment and recovery conditions, and there was a clear improvement in cell viability, reaching the values of 91% and 84% cell viability after 6 and 12 h, respectively.

In Appendix A, representative light microscopy images are presented regarding the morphology of unexposed (Appendix A) and exposed to ZnO NPs cells (Appendix A). In detail, at the end of both incubation times with ZnO NPs, there was a decrease in the number of adherent cells. When evaluating the amount of space available between the cells, the confluence was much lower relative to the control, from the end of the exposure times until the third day of recovery (Appendix A). Dark dots can be seen adhered onto the cell surface and the bottom of the cell culture plates. These dark dots are ZnO NPs, which remained strongly adhered even after the washing steps with PBS, which can be confirmed by their presence in the images captured at the 24 h recovery day (Appendix A). Over the recovery period, ZnO NPs became less visible in the available space between cells and their surface. However, at the end of the 4-day recovery period (Appendix A), due to the high confluency, cells were able to proliferate, despite the presence of ZnO NPs.

### 3.3. Effect of Chalcone ***1*** on GC-1 Cells Survival

Before assessing the protective potential of chalcone **1** against the DNA damage induced by ZnO NPs, the effect of this compound on cell viability was assessed by incubating GC-1 cells with different concentrations of chalcone **1** (0–25 µM) for 7 and 13 h. These two exposure periods were selected according to the total time that GC-1 cells would be incubated with chalcone **1** (1 h of pre-treatment followed by 6 and 12 h of co-exposure with ZnO NPs) (Appendix A).

The results presented in Figure 4 show that the tested concentrations of chalcone **1** are not toxic to GC-1 cells. Compared to the control, cells that were incubated with 3.1 µM of chalcone **1** for 7 h showed a significant increase in viability of 21.8% (*p* < 0.01). After 13 h of incubation, 3.1 µM and 12.5 µM of chalcone **1** showed similar results, with an increase in viability of 20.7% and 20.0%, respectively. In fact, all the tested concentrations resulted in higher viability levels than the control condition, except for the highest concentration (25 µM). Since 25 µM of chalcone **1** presented a tendency to decrease the viability of GC-1 cells, especially for longer incubation periods (13 h), this concentration was not considered in the subsequent assays.

Next, to examine the potential of chalcone **1** in reducing the cytotoxicity caused by ZnO NPs, GC-1 cells were pre-treated with various concentrations (1.6, 3.1, 6.25, and 12.5 µM) of this chalcone for 1 h. Then, the treatment solution with chalcone **1** was removed, and a new culture medium containing both chalcone **1** (different concentrations) and ZnO NPs was added.

As shown in Figure 5, the viability of the GC-1 cells was 85.7% and 69.6% when exposed to ZnO NPs for 6 and 12 h, respectively. However, the survival rate of cells pre-treated for 1 h with different concentrations of chalcone **1** before exposure to ZnO NPs seems to increase. In fact, under the 6 h exposure condition, all cells pre-treated with chalcone **1** had cell viability above 90%, very close to the control condition. The highest cell viability value for cells co-exposed with ZnO NPs and chalcone **1** for 6 h was 96.5% (3.1 µM chalcone **1** + NPs group), but this value was not significantly higher than the group of cells treated only with ZnO NPs (Figure 5A). However, under the 12 h treatment condition, a cell viability of 91.3% in cells co-exposed to ZnO NPs and 12.5 µM of chalcone **1** was significantly higher than that in the ZnO NPs treated group (*p* < 0.001) (Figure 5B).

### 3.4. Counteracting DNA Damage Using Chalcone ***1***

The previous work by our team indicated that the exposure of GC-1 cells to 20 µg/mL of ZnO NPs for 6 and 12 h induced DNA damage and consequently, genotoxicity [27]. To assess the potential mechanisms underlying the effect of chalcone **1** on ZnO NPs-induced genotoxicity, the presence of DNA damage was determined by measuring the intracellular levels of γ-H2AX (Ser139) by immunoblotting, which is a marker that reveals the occurrence of double-stranded breaks (DSBs) [43].

The effect of different concentrations of chalcone **1** (0–12.5 µM) on DNA integrity was tested for 7 and 13 h to assess the impact of this compound on the DNA, without the interference of NPs (Figure 6). After 7 h of exposure, none of the tested concentrations induced DNA damage, since the levels of γ-H2AX remained below the control group levels (Figure 6A). However, after 13 h of incubation, higher concentrations of chalcone **1** (6.25 and 12.5 µM) seem to increase DNA damage (Figure 6B). Compared to the control group, DNA damage increased in both timepoints after exposure to 20 µg/mL of ZnO NPs (Figure 6). When the GC-1 cells were incubated with ZnO NPs in the presence of different concentrations of chalcone **1** for 6 h, the levels of DNA damage remained higher than in the control group, but lower than in the ZnO NPs-treated group, indicating an important positive role in counteracting the DNA damage induced by ZnO NPs. This was not the case for the 12 h timepoint, since the levels of DNA damage of the co-exposed groups were higher than those in the group treated with ZnO NPs only.

### 3.5. Analysis of Cytoskeleton Dynamics

Many key spermatogenic processes, such as mitotic and meiotic divisions, rely on cytoskeleton dynamics. However, previous studies reported that ZnO NPs lead to alterations in the cytoskeletal protein of keratinocytes, epithelial cells [44], and spermatogonia [1]. ZnO NPs lead to the overproduction of ROS, which can influence the cytoskeleton through the oxidation of cytoskeletal proteins [45]. Therefore, immunoblotting analysis was performed to determine whether a compound (chalcone **1**) with antioxidant activity could protect cytoskeletal proteins from oxidative stress during exposure to ZnO NPs (Figure 7). Exposure of GC-1 cells to ZnO NPs for 6 and 12 h resulted in an increase in acetylated α-tubulin, which was significantly higher than in the control (*p* < 0.001 in both timepoints). However, increasing concentrations of chalcone **1** resulted in lower levels of this protein at both timepoints (Figure 7A,B). Co-exposure of ZnO NPs and 12.5 µM of chalcone **1** for 12 h significantly decreased acetylated α-tubulin levels, which approached the acetylated α-tubulin protein levels of those in the control group (*p* < 0.01).

β-actin levels remained unaffected by ZnO NPs exposure and chalcone **1** supplementation. There were some slight variations in the intracellular levels of this protein, but no clear trend and no statistical significance (Figure 7C,D).

## 4. Discussion

Many studies on the reproductive toxicity of NPs focus on the consequences of their exposure over time. However, uninterrupted exposure to NPs seems unrealistic under real-life conditions. Therefore, it is crucial to investigate the cell recoverability and potential consequences of NPs removal. Changes in sperm number, quality, and DNA damage induced by NPs can be reversed after the removal of NPs through treatment (e.g., SiO_2_ and Ag NPs, carbon nanotubes) [46,47,48]. This can be attributed to the fact that the toxicity of NPs causes cells to activate molecular responses to deal with the generation of oxidative stress, DNA damage, inhibition of cell division, etc. [49]. Semen quality, among other parameters, was evaluated in rabbits supplemented with ZnO-NPs and thyme oil, or their combination, demonstrating an improvement in all assays [50]. Very recently, Eman T. Hamam et al. (2022) reported positive effects of ZnO NPs on cisplatin-induced spermatogenesis impairment in rats by decreasing ROS, which in turn recovered the BTB proteins, promoted the organization of testis structure, and enhanced sperm DNA integrity [51]. However, studies pointing towards the recovery of reproductive cells after their exposure to NPs are still very scarce. Several studies have reported the adverse effects of ZnO NPs on the male reproductive system [1,52]. In fact, the present study is an extension of previous work that found that ZnO NPs had dose- and time-dependent cytotoxic effects on spermatogonia cells (GC-1) [1]. Considering that cells have a wide range of defense mechanisms, this work intended to determine to what extent GC-1 cells recover after cytotoxic exposure to ZnO NPs. GC-1 cells were exposed to 20 µg/mL of ZnO NPs for short periods (6 and 12 h). Then, cells were allowed to recover from stress in a NPs-free condition for 4 days. Different recovery periods were tested (data not shown); however, it was concluded that 4 days was the required time for cells to proliferate and reach maximum confluence. Ultimately, a comparison was made between cells immediately after treatment with ZnO NPs and cells at the end of the recovery period (Figure 3). Light microscopy images, obtained every day throughout the 4-day period, suggest a gradual recovery of cells, since they were able to divide and grow with normal epithelial morphology. On day 4, cells reach a maximum confluence, which is a good indication of the healthy state of the cells (Appendix A). Finally, a cell viability assay was also performed since it provides information about the overall health of the cells. From the results reported in Figure 3, there is a clear increase in cell viability at the end of the recovery phase. In fact, cells exposed to ZnO NPs for 6 h had a viability of 94%, very close to the control condition (Figure 3). Those that were exposed for 12 h only reached viability values of 84%. Although this percentage is lower than that achieved by the 6 h condition, there was a 25% increase in viability, higher than the 16% increase reported for the previously mentioned condition.

The full recovery (100% cell viability) of the GC-1 cells was not achieved, but very close values of 91% and 84% cell viability after 6 and 12 h, respectively, were obtained upon ZnO NPs removal (Figure 3). We speculate that these results are probably due to the NPs being firmly attached onto the cell surfaces and to the bottom of the cell culture plate, even after several PBS washing steps. Since the threat is still present, the cells’ defense system may become overburdened. However, this exposure beyond the stipulated 6 and 12 h does not seem to have resulted in greater cell damage, since cell viability increased over time. This means that cells may have adapted to the presence of NPs, which was the case for fibroblasts that were continuously exposed to 0.1 nM of gold NPs for 20 weeks. This long-term exposure changed the fibroblasts at the molecular level to reach an adapted state (new homeostasis) to the continuous stress induced by NPs to ensure survival [53].

From the close-ups of Appendix A, it is possible to observe a decrease in the number of NPs adsorbed at the bottom of the culture plate over the time of recovery. Since in the present study, NP internalization assays were not performed, one can only hypothesize about what might have caused this decrease. Over time, NPs may have detached from the bottom of the culture plates and remained floating in the culture medium. However, it was previously reported that NPs are removed from the culture environment because cells internalize them by endocytosis. One study suggested that after their uptake, cells pass NPs to their daughter cells, and if no more NPs are added into the cell culture, the concentration of ZnO NPs inside the cell becomes increasingly diluted [54]. Other studies have reported that, after their uptake, NPs are partially degraded by the acidic pH of lysosomes and then released by exocytosis [55,56,57]. This may explain not only the decrease in NPs in the cell culture, but also the cell’s ability to return to its near normal status.

Due to the GC-1 cells’ inability to fully recover from cytotoxicity induced by the ZnO NPs, even after the stipulated 4-day recovery period, another alternative treatment was investigated. Previous research has shown that treatment with agents with antioxidant properties can reduce adverse reproductive health effects [9,58]. As a result, this study also investigated the potential ameliorating effect of synthetic chalcone **1** against ZnO NPs-induced cytotoxicity, using the same cell line.

In previous a work, among many other synthetic chalcones that were studied, chalcone **1** was one of the most active chalcones in modulating the exacerbated production of ROS during neutrophils oxidative burst [10]. This activity was attributed to the lipophilicity of this chalcone, which facilitates membrane transport and the binding of this compound to intracellular targets. Since chalcone **1** showed promising results in neutrophils, in the present study, its antioxidant activity was assessed in a different cell type, a mouse spermatogonia cell line. Human lung epithelial cells were pre-treated with resveratrol (1, 5 and 10 µM) for 1 h before being exposed to 25 µg/mL of carbon black NPs for 24 h. This pre-treatment was enough to attenuate the oxidative stress induced by these NPs [59]. Mice kidney cells were also incubated with antioxidant N-acetylcysteine (0, 250, 500 µmol/L and 2, 3 mmol/L) for 1 h before being treated with 450 µmol/L of cobalt NPs for 24 h, and this pre-treatment decreased the ROS-induced cell death caused by the NPs [60]. Similar results were reported in rat neuronal cells that were pre-treated with Vitamin E (0.01–2 mM) for 1 h before being exposed to single-walled carbon nanotubes (50 µg/mL) for 24 and 48 h [61]. Hence, in the current study, a pre-treatment duration of 1 h was selected in accordance with the mentioned studies, since exposure to antioxidant agents for 1 h seems to be enough to protect cells from NP toxicity.

Therefore, GC-1 cells were pre-treated with different concentrations of chalcone **1** (0–12.5 µM) 1 h before being co-exposed to ZnO NPs and chalcone **1** for 6 and 12 h. Data from the viability assay confirmed the results obtained from other studies, indicating that the cytotoxicity of ZnO NPs is time dependent [1,2,62]. The resazurin assay revealed that incubation with ZnO NPs for 6 and 12 h resulted in cell viabilities of 85.7% and 69.6%, respectively. The results also suggest that treating cells with chalcone **1** for 1 h before the co-exposure of ZnO NPs with chalcone **1** could effectively attenuate the cytotoxicity of NPs. Pre-treatment with chalcone **1** and the combined exposure of ZnO NPs with 3.1 µM of chalcone **1** increased cell viability to 96.5% at the 6 h timepoint (Figure 5A). This means that chalcone **1** supplementation resulted in a remarkable cell viability increase of 10.8%. For the 12 h timepoint, supplementation with 12.5 µM of chalcone **1** resulted in a maximum cell viability of 91.3%, which translates into a 21.7% increase when compared to cells treated with ZnO NPs alone (*p* < 0.001) (Figure 5B). As expected, DNA damage levels increased at both timepoints when cells were exposed to ZnO NPs alone. Co-exposure with chalcone **1** consistently reduced the DNA damage at the 6 h timepoint, for all tested concentrations (Figure 6A). However, chalcone **1** seems unable to protect cells from NPs-induced genotoxicity for longer exposure times, since the levels of DNA damage elicited by ZnO NPs were not reversed by chalcone **1** supplementation at the 12 h timepoint (Figure 6B). By now, it has been well established that oxidative stress is an important source of DNA damage since the overproduction of ROS results in strand breaks and base oxidation [62]. From these results, it can also be speculated that for longer exposure periods and higher concentrations of chalcone **1**, this compound may damage cells. This would account for the fact that more DNA damage occurs when cells are co-exposed to ZnO NPs + chalcone **1** (6.25 and 12.5 µM), than when cells are exposed to ZnO NPs alone. Perhaps, for longer exposure times, higher concentrations of chalcone **1** contribute to DNA damage.

In fact, the induced DNA damage did not always correspond to the viability of GC-**1** cells since cells are able to efficiently repair their own damage. At the 12 h timepoint, the cytotoxic results contrasted with the genotoxic profile of the investigated NPs with chalcone **1**. Even though all tested concentrations of chalcone **1** with ZnO NPs resulted in higher cell viability than when cells were incubated with ZnO NPs alone, all the tested conditions exhibited more DNA damage than did the ZnO NPs group. In support of these data, Calarco et al. also reported that Polyethylenimine-PLGA NPs triggered oxidative stress and DNA damage without affecting the viability of human primary cells [63]. Human endothelial cells exposed to different concentrations of four different types of modified NPs maintained their viability, although with high levels of DNA damage and oxidative stress [64]. Therefore, it can be concluded that NPs can reveal cytotoxic properties, even though they do not always reduce cell viability.

The cytoskeleton of spermatogonia is fundamental for the initial establishment of spermatogenesis [65]. Cells can remodel their cytoskeleton in response to internal and external cues, and this ability is required for dynamic processes such as spermatogenesis because it requires extensive changes in cell shape, size, and germ cell movement [66]. The cytoskeleton comprises microfilaments, intermediate filaments, and microtubule networks. Since the role of intermediate filaments in spermatogenesis is less understood [67], in this study, only microfilaments and microtubule proteins were studied. Microfilaments are solid rods made of actin. Microtubules are cylindrical polymers formed by α- and β-tubulin heterodimers, but α-tubulin undergoes acetylation in the cytoplasm, which provides flexibility, mechanical resistance, and overall stability to the microtubules [68]. However, NPs can interfere with this dynamic and lead to the breakdown and reorganization of the cytoskeleton [69], which can affect many cellular processes essential for spermatogenesis.

Immunoblotting analysis reveals a clear increase in acetylated α-tubulin levels after GC-1 cells exposure to ZnO NPs for 6 and 12 h (Figure 7A,B). Alterations in the microtubule network in the presence of NPs are not always related to lower levels of α-tubulin acetylation [70]. Instead, this increase may be a cell coping mechanism to make microtubules softer and thus, more resistant to damage induced by mechanical bending [71]. However, these levels decreased and approached normal values as the concentration of chalcone **1** increased, suggesting that this compound may have exerted protective effects. The reported alterations in the cytoskeleton could have been a result of the increased levels of ROS, a product of mechanical stress from the direct interaction ZnO NPs with the cytoskeleton [72] or the result of the dissolution of zinc ions [44]. Tubulin molecules contain many zinc-binding sites and act as zinc scavengers. Therefore, it is possible that zinc ions released from ZnO NPs may have triggered tubulin polymerization and the formation of large, aberrant microtubules [44].

Most studies that assessed the impact of NPs on cytoskeletal integrity have found a destabilization and degradation of actin filaments [73]. However, in the present work, β-actin levels remained unaltered under all treatment conditions at both timepoints. There were only slight variations, without statistical significance (Figure 7C, D), which suggests that β-actin remained unaffected, probably because short exposure times were used.

## 5. Conclusions and Future Trends

In this study, ZnO NPs at 20 µg/mL affected the viability of GC-1 cells, especially for longer incubation times (12 h). After this exposure, the question of whether the adverse effects caused by ZnO NPs on GC-1 cells were reversible was addressed. The results show that the viability of GC-1 cells increased after 4 days of proliferation in an NP-free environment; however, the results also suggest that these cells were not able to fully recover. Thus, to support the cells’ defense system and promote their recovery, a synthetic chalcone with antioxidant potential was synthesized and used during the exposure to ZnO NPs. The results from this second approach suggest that chalcone **1** can partially recover the cell viability of the GC-1 cells, appearing to counteract DNA damage when cells are exposed to high concentrations of ZnO NPs for 6 h. Additionally, chalcone **1** may have some protective effects on the cytoskeleton dynamics through its stabilization. However, when GC-1 cells are exposed to high concentrations of ZnO NPs for longer periods (12 h), the results indicate some cell toxicity, especially when combined with the higher doses of chalcone **1**. Of note, this compound attenuates the NPs-induced genotoxicity for shorter exposure times. However, more research is needed to explain the precise behavioral mechanisms of this compound in the presence of ZnO NPs.

## Figures and Tables

**Figure 1 nanomaterials-12-03561-f001:**
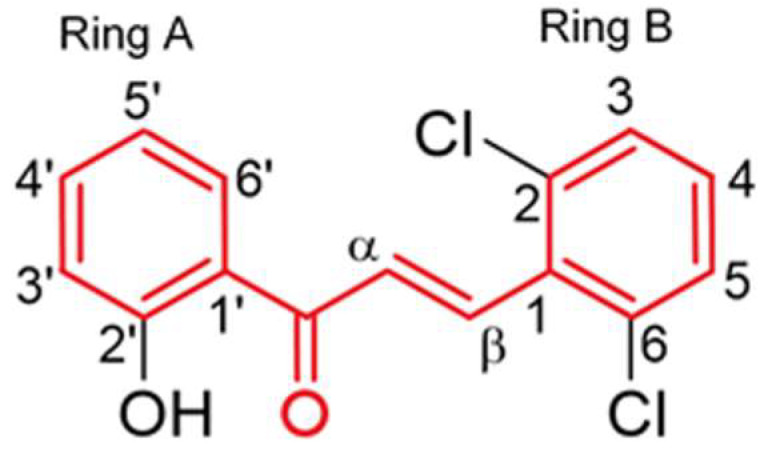
Structure of the synthesized chalcone **1** [(*E*)-3-(2,6-dichlorophenyl)-1-(2-hydroxyphenyl)prop-2-en-1-one]. The 1,3-diarylprop-2-en-1-one fragment is marked in red.

**Figure 2 nanomaterials-12-03561-f002:**
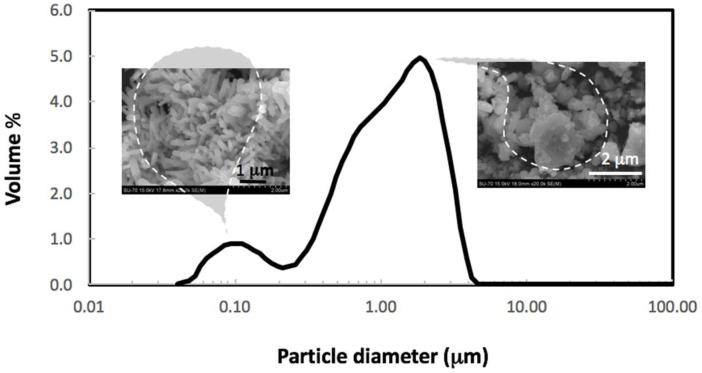
Particle size distribution and SEM images of the ZnO powder used.

**Figure 3 nanomaterials-12-03561-f003:**
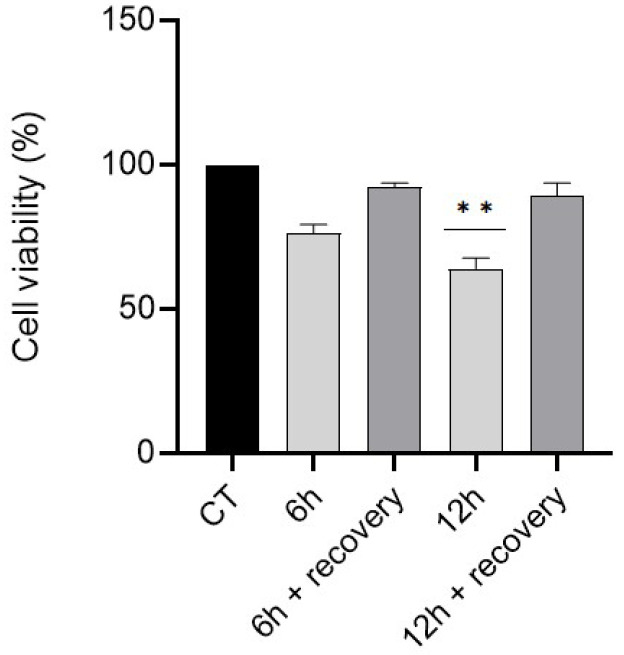
**GC-1 cells recovery upon NPs exposure.** Results from viability assay analysis of GC-1 cells after their incubation with 20 µg/mL of ZnO NPs, for 6 and 12 h, and after a recovery period of 4 days. The percentage of viable cells for each condition was plotted as the mean ± SEM of four independent experiments. Values are expressed as arbitrary units, and the cell viability of the control condition was given a value of 100. ** *p* < 0.01 compared to the control group, using one way ANOVA. CT: control.

**Figure 4 nanomaterials-12-03561-f004:**
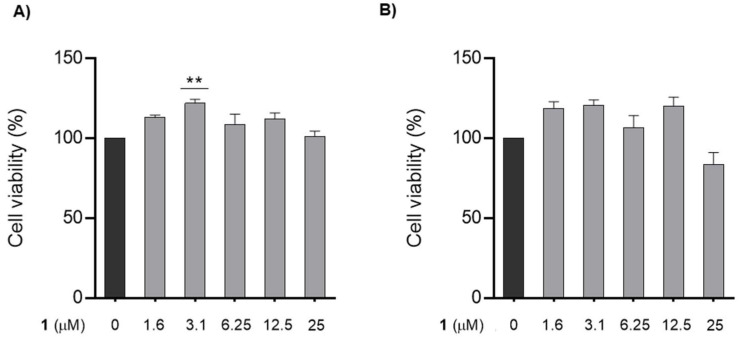
**Cell viability assay of cells treated with chalcone 1** (for the sake of simplicity, the chalcone is indicated in the graphics using only the number **1**). Changes in the viability of the GC-1 cells after being treated with different concentrations of chalcone **1** (0–25 µM) for (**A**) 7 h and (**B**) 13 h. The percentage of viable cells for each condition was plotted as mean ± SEM of five independent experiments, performed in duplicate. Values are expressed as arbitrary units, and the cell viability of the control condition is given a value of 100. ** *p* < 0.01 compared to the control group, using a nonparametric Kruskal–Wallis one-way ANOVA test.

**Figure 5 nanomaterials-12-03561-f005:**
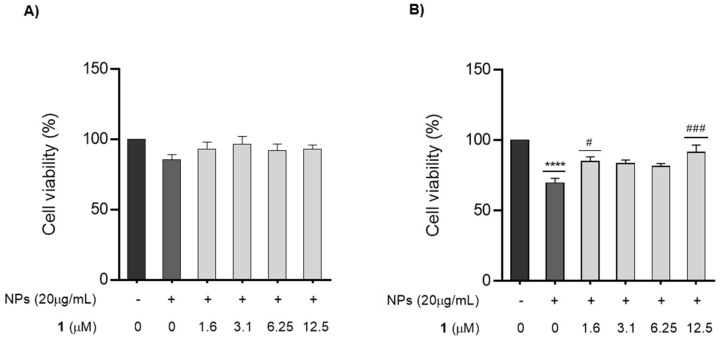
**Cell viability assay of cells co-exposed with chalcone 1 and ZnO NPs.** Changes in the viability of the GC-1 cells treated with 20 µg/mL of ZnO NPs in the absence and presence of different concentrations of chalcone **1** (0–12.5 µM) after (**A**) 6 and (**B**) 12 h of exposure. The percentage of viable cells for each condition was plotted as mean ± SEM of five independent experiments, performed in duplicate. Values are expressed as arbitrary units, and the cell viability of the control condition is given a value of 100. **** *p* < 0.0001 compared to the control group, # *p* < 0.05 and ### *p* < 0.001 compared to the group treated with ZnO NPs, using one-way ANOVA. NPs: nanoparticles.

**Figure 6 nanomaterials-12-03561-f006:**
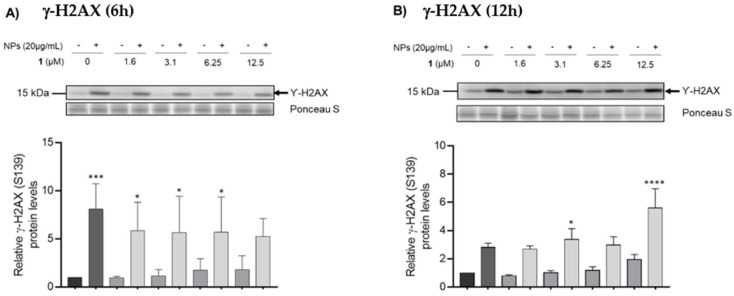
**Immunoblotting analysis of γ-H2AX (Ser139) intracellular levels**. This assay allows for the assessment of the DNA damage after the incubation of GC-1 cells with 20 µg/mL of ZnO NPs and different concentrations of chalcone **1** (0–12.5 µM) per se, and the protective effects of chalcone **1** against the genotoxicity induced by the NPs after (**A**) 6 h and (**B**) 12 h of co-exposure with ZnO NPs. The intracellular protein levels of γ-H2AX were estimated in relation to protein levels detected in the control condition. Ponceau S staining was used to assess gel loading. * *p* < 0.05, *** *p* < 0.001, and **** *p* < 0.0001 compared to the control group, using one-way ANOVA. All data were expressed as the mean ± SEM of four independent experiments. NPs: nanoparticles.

**Figure 7 nanomaterials-12-03561-f007:**
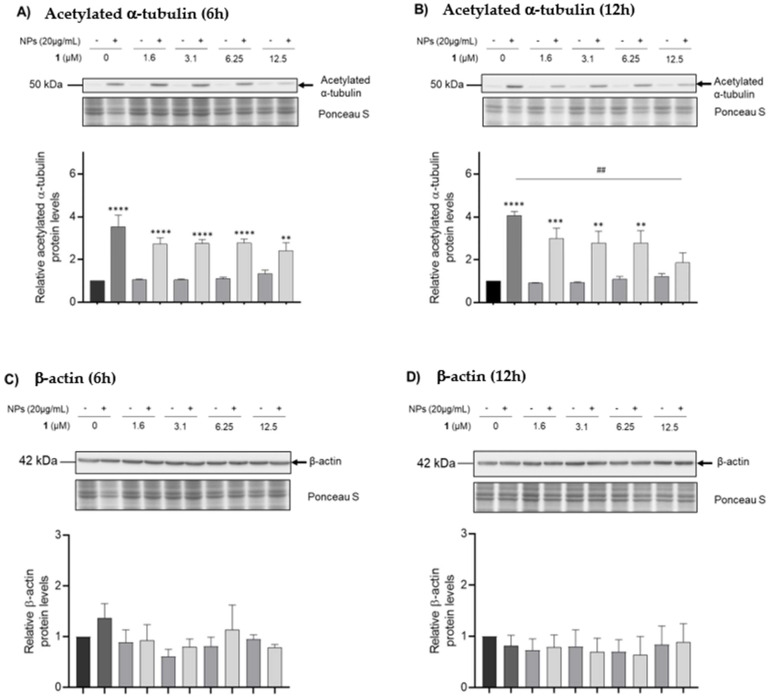
**Immunoblotting analysis of cytoskeleton proteins.** Acetylated α-tubulin intracellular protein levels of GC-1 cells treated with ZnO NPs (20 µg/mL) and chalcone **1** (0–12.5 µM) for 6 (**A**) and 12 h (**B**). The intracellular protein levels of acetylated α-tubulin were estimated in relation to protein levels detected in the control condition. Ponceau S staining was used to assess gel loading. All data were expressed as the mean ± SEM of four independent experiments. ** *p* < 0.01, *** *p* < 0.001 and, **** *p* < 0.0001 compared to the control group, using one-way ANOVA, and ## *p* < 0.01 compared to the group treated with ZnO NPs using the Student’s *t*-test. β-actin intracellular protein levels of GC-1 cells treated with ZnO NPs (20 µg/mL) and chalcone **1** (0–12.5 µM) for 6 (**C**) and 12 h (**D**). The intracellular protein levels of β-actin were estimated in relation to protein levels detected in the control condition. Ponceau S staining was used to assess gel loading. All data were expressed as mean ± SEM of three independent experiments. No significant differences were detected between groups when analyzed by one-way ANOVA. NPs: nanoparticles.

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
