# Peer review of "Different Strategies to Attenuate the Toxic Effects of Zinc Oxide Nanoparticles on Spermatogonia Cells"

_nanomaterials, 2022, doi:10.3390/nano12203561_

Round 1

Reviewer 1 Report

The authors present a work on the toxicity of zinc nanoparticles on sperm cells and the effect of the antioxidant chalcone. The compostion of experiments and measured markers for adverse effect seem to be randomly selected. I do not see a recurrent theme that analyses the before mentioned topic. I think the authors should focus on one specific topic like the connection of DANN damage and cell viability, possible long term effects of zinc nanoparticles for example. To observe recovery of cells at one specific time point of a cell culture also seems to be very random and i do not see the impact of the results from this one time point. In the present form, i do not see the merit of the manuscript to the topic of zinc nanoparticle indued toxicity and therefore suggest to reject the present article for publication.

Specific comments to the manuscript can be found in the following:

Line 27 please review the grammar of this sentence

Line 28 please review the grammar of this sentence

Line 278/279 delete sentence These values….. it is dispensible

286 please review the word order

291 please specify the reason for stating this. Since you showed above that cells died upon exposure, of course the confluency was decreased since dead cells detach from the surface

Figure axes: please add if relative or normalized to control values are shown.

Please add what is shown in Figure A and Figure B in the text.

I don‘t see the point why you only use 1 when referring to chalcone. It would be better to state at the beginning that when using chalcone you refer to chalcone 1. But using only 1 interferes with fluent reading.

Line 426 i dont get what exactly is meant by full recovery was not possible. Did you check the cell number after more than 4 days of recovery?

Line 435 a decrease compared to what?

Conclusions: chalcone 1 induced DNA damage  and should definitely be mentioned as one important conclusion to be careful when applying chalcone.

Author Response

Reviewer 1:

The authors present a work on the toxicity of zinc nanoparticles on sperm cells and the effect of the antioxidant chalcone. The compostion of experiments and measured markers for adverse effect seem to be randomly selected. I do not see a recurrent theme that analyses the before mentioned topic. I think the authors should focus on one specific topic like the connection of DANN damage and cell viability, possible long term effects of zinc nanoparticles for example. To observe recovery of cells at one specific time point of a cell culture also seems to be very random and i do not see the impact of the results from this one time point. In the present form, i do not see the merit of the manuscript to the topic of zinc nanoparticle indued toxicity and therefore suggest to reject the present article for publication.

Authors R: Thank you for your overall positive and relevant comments that received our major attention. Several changes in the manuscript were performed according to your suggestions. The manuscript was also revised by an English native speaker.

Specific comments to the manuscript can be found in the following:

Reviewer 1. Line 27 please review the grammar of this sentence

Authors R: The sentence was changed. ‘Additionally, the impact of a synthetic chalcone (E)-3-(2,6-dichlorophenyl)-1-(2-hydroxyphenyl)prop-2-en-1-one (1) to counteract the reproductive toxicity of ZnO NPs was investigated’.

Reviewer 1. Line 28 please review the grammar of this sentence

Authors R: Thank you the reviewer, this sentence was deleted.

Reviewer 1. Line 278/279 delete sentence These values….. it is dispensible

Authors R:  As suggested by the reviewer, the following sentence was removed ‘These values are very close to 100%’

Reviewer 1. 286 please review the word order

Authors R: The sentence was changed as follows: ‘In supplementary Figures 6 and 7, representative light microscopy images are presented regarding the morphology of unexposed (supplementary Figures 6 and 7, A-E) and exposed to ZnO NPs cells (supplementary Figures 6 and 7, F-J).’

Reviewer 1. 291 please specify the reason for stating this. Since you showed above that cells died upon exposure, of course the confluency was decreased since dead cells detach from the surface

Authors R: We understand the issue raised by the reviewer. The following sentence was just to indicate that cells are less confluent than controls until the third day of recovery. In fact, in third and fourth day of recovery they become almost confluent.

‘When evaluating the amount of space available between cells, the confluence was much lower relative to the control, from the end of the exposure times until the third day of recovery (supplementary Figures 6 and 7, F-I).’

Reviewer 1. Figure axes: please add if relative or normalized to control values are shown.

Authors R: Thank you for the highly relevant comment. We added the missing relevant information in Figure 3, 4, 5, 6 and 7 legends.

Reviewer 1. Please add what is shown in Figure A and Figure B in the text.

Authors R: Thank you for your suggestion. It was added all over the manuscript.

Reviewer 1. I don‘t see the point why you only use 1 when referring to chalcone. It would be better to state at the beginning that when using chalcone you refer to chalcone 1. But using only 1 interferes with fluent reading.

Authors R: We understand the reviewer comment. However, when dealing with organic compounds, it is very common to give an arabic number to the compound structure which is indicated using bold style to not interfere with reading.

In fact, several examples of this widely adopted way of referring to an organic compound in the text can be found in the literature (e.g 10.1016/j.ijbiomac.2021.04.061; 10.1021/acs.jnatprod.0c00728; 10.3390/nu14020306; 10.3390/molecules27196169). In the present manuscript and following the reviewer suggestion, most of the times we used chalcone 1 instead of 1 along the text, and 1 (in bold style) in the graphics for sake of simplicity.

Reviewer 1. Line 426 i dont get what exactly is meant by full recovery was not possible. Did you check the cell number after more than 4 days of recovery?

Authors R: We understand the reviewer concern. We did experiments longer than 4 days of recovery, with 5 and 6 days of recovery and the results were quite similar. The cells never reach the 100% of cell viability (full recovery). We speculate that these results are probably due to the NPs firmly attached onto cell surfaces and to the bottom of the cell culture plate, even after several washing steps with PBS. These NPs affect cells viability.

To clarify, the following sentence was added in the manuscript:

‘GC-1 cells full recovery (100 % cell viability) was not achieved, but very close values of 91% and 84% cell viability within 6 and 12h, respectively were obtained upon ZnO NPs removal (Figure 3). We speculate that these results are probably due to the NPs firmly attached onto cell surfaces and to the bottom of the cell culture plate, even after several washing steps with PBS’.

Reviewer 1. Line 435 a decrease compared to what?

Authors R: The number of NPs adsorbed at the bottom of the culture plate decreased all over the time of recovery as observed in supplementary Figures 6 and 7 (F-H). To clarify this issue, the sentence was changed as follows:

‘From the close-ups of supplementary Figures 6 and 7 (F-H), it is possible to observe a decrease in the number of NPs adsorbed at the bottom of the culture plate, over the time of recovery’.

Reviewer 1. Conclusions: chalcone 1 induced DNA damage and should definitely be mentioned as one important conclusion to be careful when applying chalcone.

Authors R: Thank you for the suggestion. Chalcone seems to counteract DNA damage when cells are exposed to high concentrations of ZnO NPs for short periods (6h). However, this is not true for longer periods (12 h).

We decided to rephrase several sentences in conclusion, to make it clear.

‘The results from this second approach suggest that chalcone 1 can partially recover the cell viability of GC-1 cells and seems to counteract DNA damage when cells are exposed to high concentrations of ZnO NPs for 6h. Additionally, chalcone 1 may have some protective effects on the cytoskeleton dynamics through its stabilization. However, when GC-1 cells are exposed to high concentrations of ZnO NPs for longer periods (12h) the results indicate some cell toxicity, especially when combined with the higher doses of chalcone 1’.

Reviewer 2 Report

The manuscript by Vassal et al reported their investigate on the recovery strategies for ZnO induced toxicity to mouse spermatogonia cells. The first strategy was to allow the cells to recover in a ZnO-free environment. Another strategy was using a synthetic chalcone, an antioxidant. The study suffers a few major flaws that should be carefully addressed before publication.

First, the justification of this study is relatively weak at this moment. The toxicity of ZnO was well-understood, by using various types of in vitro cell lines and in vivo models. The oxidative stress paradigm was well-established. Even if studies performed on a male reproductive cell line were not reported before, using a spermatogonia cell line to study the toxicity of ZnO barely makes the novelty standout, especially if the toxicity mechanism remains the same.

Second, the recover strategies applied were also problematic. Removing the exposure of ZnO could barely be called a recovery strategy. And using an antioxidant was a norm to any reagent that would cause oxidative stress. Was there any novelty involved in the creation of the antioxidant?

Some other issues,

The following sentence was too wordy and should be revised. "Additionally, an-24 other treatment strategy was used based on the use of a synthetic chalcone, the (E)-3-(2,6-dichloro-25 phenyl)-1-(2-hydroxyphenyl)prop-2-en-1-one (1) with antioxidant properties and its impact to alle-26 viate the reproductive toxicity of ZnO NPs was investigated." Similar issues were throughout the manuscript.

Figure 2 seems unnecessary. The overall experimental setup was not complicated and such diagram seems redundant.

Author Response

Reviewer 2:

The manuscript by Vassal et al reported their investigate on the recovery strategies for ZnO induced toxicity to mouse spermatogonia cells. The first strategy was to allow the cells to recover in a ZnO-free environment. Another strategy was using a synthetic chalcone, an antioxidant. The study suffers a few major flaws that should be carefully addressed before publication.

First, the justification of this study is relatively weak at this moment. The toxicity of ZnO was well-understood, by using various types of in vitro cell lines and in vivo models. The oxidative stress paradigm was well-established. Even if studies performed on a male reproductive cell line were not reported before, using a spermatogonia cell line to study the toxicity of ZnO barely makes the novelty standout, especially if the toxicity mechanism remains the same.

Authors R: We would like to thank the positive criticisms of the reviewer that will help our manuscript improvement. Regarding the justification of this study, it was clarified by introducing a new explanatory sentence.

‘Bearing in mind previous studies from our team (Pinho et al 2020), where GC-1 cells were exposed to high doses of ZnO NPs for short periods of time (6h and 12h) and cell toxicity, DNA damage and dramatic alterations in cytoskeleton and nucleoskeleton were observed, we hypothesized if the observed GC-1cells toxicity may be somehow reversed. To test this hypothesis two approaches were used. We investigated if GC-1 cells toxicity may be reversed by simply removal of the ZnO NPs and if chalcone 1 may counteract the GC-1 cells damage observed’.

Reviewer 2: Second, the recover strategies applied were also problematic. Removing the exposure of ZnO could barely be called a recovery strategy. And using an antioxidant was a norm to any reagent that would cause oxidative stress. Was there any novelty involved in the creation of the antioxidant?

Authors R: We would like to thank the positive criticisms of the reviewer that will help our manuscript improvement. Chalcone 1 is a synthetic compound that was synthesized in our laboratory and that belongs to a family of more lipophilic chlorinated chalcones whose antioxidant activity was studied in a previous work (Medicinal Chemistry, 2022, 18(1), 88-96. DOI: 10.2174/1573406417666201230093207). Among the chalcones belonging to that family, chalcone 1 was found to be the most active, in the previous work, and therefore it was selected for the present study.

Some other issues,

Reviewer 2: The following sentence was too wordy and should be revised. "Additionally, another treatment strategy was used based on the use of a synthetic chalcone, the (E)-3-(2,6-dichloro-25 phenyl)-1-(2-hydroxyphenyl)prop-2-en-1-one (1) with antioxidant properties and its impact to alleviate the reproductive toxicity of ZnO NPs was investigated." Similar issues were throughout the manuscript.

Authors R: This sentence was revised, as suggested by the reviewer as follows.

‘Additionally, the impact of a synthetic chalcone (E)-3-(2,6-dichlorophenyl)-1-(2-hydroxyphenyl)prop-2-en-1-one (1) to counteract the reproductive toxicity of ZnO NPs was investigated’.

Reviewer 2: Figure 2 seems unnecessary. The overall experimental setup was not complicated and such diagram seems redundant.

Authors R: Thank you for the reviewer suggestion. Figure 2 was removed from the main manuscript but kept as supplementary material (supplementary Figure 5).

Author Response

Reviewer 3:

The worldwide presence of ZnO nanoparticles is an important problem, especially in the context of not fully established impact on tissues and organs. The Authors addressed the topic of the effects of ZnO nanoparticles on spermatogonia which in the era of increased problems with human fertility seems to be very important. The paper is interesting, however, requires some corrections. I hope the following comments will be helpful to improve the manuscript.

Authors R: We appreciate very much your positive and constructive comments and suggestions on our manuscript.

Reviewer 3: Introduction section - line 44 - please add some examples of the ZnO NPs use and sources of exposure

Authors R: Thank you for the suggestion. The first paragraph of introduction section was revised and completed:

‘A growing amount of evidence has demonstrated that zinc oxide nanoparticles (ZnO NPs) can interfere with all types of male reproductive cells in a dose and time-dependent manner [1–3]. Due to their widespread use, in personal care products, coatings, paints, electronic circuits, sensors [4], and even in antimicrobial food packaging [5], its accidental inhalation, ingestion, or dermal absorption is almost inevitable [6]’.

References

Kalpana VN, Devi Rajeswari V. A Review on Green Synthesis, Biomedical Applications, and Toxicity Studies of ZnO NPs. Bioinorg Chem Appl. 2018 Aug 1;2018:3569758. doi: 10.1155/2018/3569758. PMID: 30154832; PMCID: PMC6093006.

Fontecha-Umaña F, Ríos-Castillo AG, Ripolles-Avila C, Rodríguez-Jerez JJ. Antimicrobial Activity and Prevention of Bacterial Biofilm Formation of Silver and Zinc Oxide Nanoparticle-Containing Polyester Surfaces at Various Concentrations for Use. Foods. 2020 Apr 6;9(4):442. doi: 10.3390/foods9040442. PMID: 32268566; PMCID: PMC7230149.

Staron, A. et al. Analysis of the Exposure of Organisms to the Action of Nanomaterials. Materials. 2020 Jan 12;13(349). doi:10.3390/ma13020349

Reviewer 3: - line 57 – please add some examples of plans containing chalcones

Authors R: Thank you the reviewer for the interesting suggestion.

Chalcones have been the subject of several review manuscripts about their natural occurrence in plants, broad and remarkable biological activities, and methods for their synthesis [1,2]. Additionally, several research articles have been published on the synthesis of novel chalcones derivatives and investigation of their pharmacological activities. Two examples, among a huge number of articles are, for example, the study developed by Rocha et al. that have synthesized a wide range of chalcone derivatives and evaluated their ability to inhibit the carbohydrate-hydrolyzing enzymes α-amylase and α-glucosidase to be used as antidiabetic agents [3], and Martins et al. who synthesized several families of chalcones and investigated their potential as scavengers of HOCl and as inhibitors of oxidative burst [4].

[1] Zsuzsanna Rozmer, Pál Perjési, Naturally occurring chalcones and their biological activities. Phytochem Rev. 2016, 15, 87–120 (2016). DOI: 10.1007/s11101-014-9387-8

[2] Chunlin Zhuang, Wen Zhang, Chunquan Sheng, Wannian Zhang, Chengguo Xing, Zhenyuan Miao. Chalcone: A Privileged Structure in Medicinal Chemistry. Chem. Rev. 2017, 117, 12, 7762–7810. DOI: 10.1021/acs.chemrev.7b00020

[3] Sónia Rocha, Adelaide Sousa, Daniela Ribeiro, Catarina M. Correia, Vera L. M. Silva,  Clementina M. M. Santos, Artur M. S. Silva,  Alberto N. Araújo, Eduarda Fernandes  and Marisa Freitas. A study towards drug discovery for the management of type 2 diabetes mellitus through inhibition of the carbohydrate-hydrolyzing enzymes α-amylase and α-glucosidase by chalcone derivatives. Food Funct., 2019, 10, 5510, DOI: 10.1039/c9fo01298b

[4] Thaise Martins, Vera L. M. Silva, Artur M. S. Silva, José L. F. C. Lima, Eduarda Fernandes and Daniela Ribeiro, Chalcones as scavengers of HOCl and inhibitors of oxidative burst: Structure-activity relationship studies. Medicinal Chemistry, 2022, 18(1), 88-96. DOI: 10.2174/1573406417666201230093207

Reviewer 3: Materials and Methods section

- please explain shortly how the chalcone 1 concentrations were chosen

Authors R: The chalcone 1 concentrations were chosen based on previous studies with this type of compounds as inhibitors of oxidative burst (Medicinal Chemistry, 2022, 18(1), 88-96. DOI: 10.2174/1573406417666201230093207). For further information, please see the concentrations used for chalcone 4b (same structure as chalcone 1) in the referred study.

This sentence was added in materials and methods section.

Reviewer 3:- in general, the abbreviation “1” for chalcone 1 used in the study is not the best

– it is easily “missed” during reading

– I suggest using e.g., CHAL1 or something similar

Authors R: Thanks for the comment. When dealing with organic compounds, it is very common to give an arabic number to the compound structure which is indicated using bold style to not interfere with reading. In fact, several examples of this widely adopted way of referring to an organic compound in the text can be found in the literature (e.g 10.1016/j.ijbiomac.2021.04.061; 10.1021/acs.jnatprod.0c00728; 10.3390/nu14020306; 10.3390/molecules27196169).

In the present manuscript, to avoid confusion and following this reviewer suggestion, most of the times we used chalcone 1 instead of 1 along the text, and 1 (in bold style) in the graphics for sake of simplicity.

Reviewer 3: - it is not very clear what is the difference between the second experimental strategy I) and II) in the context of exposure to ZnO NPs

– is it a difference in the analyses performed? – please explain it in more detail

Authors R: The difference between the second experimental strategy I) and II) is that in the first, GC-1 cells were incubated only with several concentrations of chalcone 1 and the cell viability monitored. In the second, cells were exposed to 20 mg/ml of ZnO NPs for 6 and 12 h in the absence and presence of different concentrations of chalcone 1 (0-12.5 mM). To clarify this issue, the legend of supplementary figure 5 was improved as follows.

‘First, (I) cells were incubated only with different chalcone 1 concentrations (0-25 µM) for 7 and 13 h to test the cytotoxicity of the compound. Then, cells were exposed to chalcone 1 (0-12.5 µM) or 20 µg/mL of ZnO NPs (II) in the absence and presence of different concentration of chalcone 1 (0-12.5 µM). III) Cells were pretreated with chalcone 1 for 1 h and co-exposed to ZnO NPs and chalcone 1 to evaluate the protective effects of chalcone 1 against ZnO NPs toxicity. IB, immunoblotting; NPs, nanoparticles.’

Reviewer 3: Results section - it would be good to add more photos from cell cultures from the second experimental strategy with chalcone 1 (in supplementary materials) as it is done for the first experimental strategy

Authors R: We thank the reviewer suggestion, in fact, we carefully monitored the cells during second experimental strategy with chalcone 1, but unfortunately, we don´t have photos to all conditions tested.

Reviewer 3:  line 266 – “[20]”

Authors R: Very sorry about this mistake, which was already corrected.

Reviewer 3: – please correct the reference - line 278 – “within 6 and 12h, respectively” – shouldn’t it be “after 6 and 12h exposure, respectively” – the recovery was 4 days, not 6 or 12 hours - Figure 7 A-B and Figure 8 A-D – please add group description in the graphs

Authors R: Thank you the reviewer for the correction. We also added the information in Figures 6 and 7 (previous Figures 7 and 8).

Reviewer 3: Discussion section - please consider extending a discussion with consideration about postulated positive effect of ZnO NPs in the male reproductive system (e.g., Hamam et al., 2022, doi: 10.1016/j.tox.2022.153102; Abdel-Wareth et al., 2020, doi: 10.3390/ani10122234 or ElMaddawy and Abd El Naby, 2019, doi: 10.1039/c9tx00052f)

Authors R: Thank you. The positive effects of ZnO NPs were mentioned. Then, these papers were added, at discussion section as follows.

‘Semen quality, among other parameters, were evaluated on rabbits supplemented with ZnO-NPs and thyme oil or their combination, demonstrating an improvement in all assays [48]. Very recently, Eman T. Hamam and co-authors (2022) reported positive effects of ZnO NPs on cisplatin-induced spermatogenesis impairment in rats by decreasing ROS, which in turn recovered the BTB proteins, promoted the organization of testis structure and enhanced sperm DNA integrity [49]. However, studies pointing towards the recovery of reproductive cells after their exposure to NPs are still very scarce.

References

Abdel-Wareth AAA, Al-Kahtani MA, Alsyaad KM, Shalaby FM, Saadeldin IM, Alshammari FA, Mobashar M, Suleiman MHA, Ali AHH, Taqi MO, El-Sayed HGM, El-Sadek MSA, Metwally AE, Ahmed AE. Combined Supplementation of Nano-Zinc Oxide and Thyme Oil Improves the Nutrient Digestibility and Reproductive Fertility in the Male Californian Rabbits. Animals (Basel). 2020 Nov 27;10(12):2234. doi: 10.3390/ani10122234. PMID: 33261201; PMCID: PMC7761441.

Hamam ET, Awadalla A, Shokeir AA, Aboul-Naga AM. Zinc oxide nanoparticles attenuate prepubertal exposure to cisplatin- induced testicular toxicity and spermatogenesis impairment in rats. Toxicology. 2022 Feb 28;468:153102. doi: 10.1016/j.tox.2022.153102. Epub 2022 Jan 21. PMID: 35074511.

Reviewer 3: References section - please check and correct references according to the requirements of the Journal

Authors R: The references were corrected and formatted according to journal requirements.

Additional remarks - list of the Authors – A.M.S. Silva – please change the font - please add the Authors’ initials next to the affiliations - please add the Authors’ Contributions

Authors R: Thank you the reviewer. All these alterations were included in the manuscript.

Round 2

Reviewer 1 Report

the authors did comment on the suggestions I had. however the overall study did not change and this is why i will not change my first suggestion to reject the manuscript.

Author Response

Answers to comments of Reviewers

Reviewer 1: the authors did comment on the suggestions I had. however the overall study did not change and this is why i will not change my first suggestion to reject the manuscript.

Authors R:

  • This comment seems unfair to us, since all the suggestions and explanations asked by the reviewer were added to the manuscript. As mentioned in revision 1, the present study was based on our previous study (Pinho et al, 2020) published in the prestigious journal Cells that was well received by scientific community.

The experiments and the measured markers used in the present manuscript were not randomly selected as suggested by the reviewer 1. These were carefully chosen in our previous study (Pinho et al, 2020), where GC-1 cells were exposed to high doses of ZnO NPs for short periods (6h and 12h), and cell toxicity, DNA damage and dramatic alterations in cytoskeleton and nucleoskeleton were observed. Given these results, we hypothesized if the observed GC-1 cells toxicity may be somehow reversed. To test this hypothesis in the present study, two approaches were used: we investigated if GC-1 cells toxicity may be reversed by simply removal of the ZnO NPs, and if chalcone 1 may counteract the GC-1 cells damage observed.

Therefore, the same markers: cell viability, DNA damage and cytoskeleton dynamics were used in both studies (previous and present study), since the idea of the present study was to reverse the observed cytotoxic effects.

  • We are very sorry that the reviewer 1 was not able to understand our study, given that the major focus on the present study was precisely the connection between DNA damage and cell viability when GC-1 cells are exposed to high doses of ZnO NPS for short periods (6h and 12h). However, as it was reported in our previous study, ZnO NPs decrease cell metabolism and membrane integrity in a dose-dependent manner, which is in agreement with the increase in cell death and is associated with severe morphological changes in proteins of cytoskeleton, such as tubulin and actin. Therefore, in the present study, we also consider the evaluation of the cytoskeleton integrity and dynamics an important aspect.

  • The reviewer suggestion for studying the long-term effects of ZnO NPs, are not plausible given those are very difficult or impossible to reverse. In fact, our results clearly indicate that chalcone 1 can partially recover the cell viability of GC-1 cells and seems to counteract DNA damage when cells are exposed to high concentrations of ZnO NPs for 6h. Additionally, chalcone 1 may have some protective effects on the cytoskeleton dynamics through its stabilization. However, when GC-1 cells are exposed to high concentrations of ZnO NPs for longer periods (12h) the results indicate some cell toxicity, especially when combined with the higher doses of chalcone 1.

Reviewer 2 Report

Some language checking is needed before publication.

Author Response

Answers to comments of Reviewers

Reviewer 2: Some language checking is needed before publication.

Authors R: Thank you the reviewer for the suggestion. The present version of the manuscript was carefully read by the authors.
